# Wireless battery-free body sensor networks using near-field-enabled clothing

Rongzhou Lin[1,8]*, Han-Joon Kim [2,8], Sippanat Achavananthadith[2], Selman A. Kurt [2], Shawn C.C. Tan [3], Haicheng Yao[4], Benjamin C.K. Tee [1,4,5,6], Jason K.W. Lee [3,5,7] & John S. Ho [1,2,5,6]*

Networks of sensors placed on the skin can provide continuous measurement of human physiological signals for applications in clinical diagnostics, athletics and human-machine interfaces. Wireless and battery-free sensors are particularly desirable for reliable long-term monitoring, but current approaches for achieving this mode of operation rely on near-field technologies that require close proximity (at most a few centimetres) between each sensor and a wireless readout device. Here, we report near-field-enabled clothing capable of establishing wireless power and data connectivity between multiple distant points around the body to create a network of battery-free sensors interconnected by proximity to functional textile patterns. Using computer-controlled embroidery of conductive threads, we integrate clothing with near-field-responsive patterns that are completely fabric-based and free of fragile silicon components. We demonstrate the utility of the networked system for real-time, multi-node measurement of spinal posture as well as continuous sensing of temperature and gait during exercise.

[1] Institute for Health Innovation and Technology, National University of Singapore, Singapore 117599, Singapore. [2] Department of Electrical and Computer Engineering, National University of Singapore, Singapore 117583, Singapore. [3] Department of Physiology, Yong Loo Lin School of Medicine, National University of Singapore, Singapore 117593, Singapore. [4] Department of Materials Science and Engineering, National University of Singapore, Singapore 117575, Singapore. [5] The N.1 Institute for Health, National University of Singapore, Singapore 117456, Singapore. [6] Smart Systems Institute, National University of Singapore, Singapore 119613, Singapore. [7] Global Asia Institute, National University of Singapore, Singapore 119076, Singapore. [8] These authors contributed equally: Rongzhou Lin, Han-Joon Kim. *email: rongzhoulin@nus.edu.sg; johnho@nus.edu.sg

Wearable sensors can improve the diagnosis and treatment of many medical conditions by enabling continuous monitoring of health outside of clinical or laboratory settings. Advances in materials and electronics over the past decade have yielded a wide range of sensors that can be comfortably and reliably placed on the skin to measure temperature, electrical activity, blood oxygenation, sweat chemical composition, and other physiological parameters[1–16]. In broad applications ranging from vital signs monitoring to fitness tracking, simultaneous recording of activity at different anatomical locations can enhance the utility and reliability of the measurement modality. However, practical and convenient acquisition of spatially distributed physiological signals is challenging because of the need to interconnect multiple sensors around the body. For instance, current clinical monitoring systems rely on wires to connect sensors to a central hub for powering and data acquisition, but such tethers constrain physical motion and have limited use outside of the clinic.

Wireless technologies can be used to connect wearable sensors without physical constraints. In particular, radio-based wireless communication methods, such as Bluetooth and Wi-Fi, are widely used to enable sensors to wirelessly communicate around the body for monitoring health and providing real-time clinical notifications[11,17–24]. Unlike wired interconnects, however, these radio-based technologies require each sensor to be separately powered, typically using rigid batteries or bulky energy harvesters. These components limit the degree of skin conformability and user comfort that can be achieved, and require periodic replacement or availability of specialised energy sources for long-term function[25–28]. In addition, the radiative nature of data transmission results in vulnerabilities to eavesdropping and necessitates the use of cryptography techniques to address privacy concerns[15,16,24,29,30]. Near-field communication (NFC) is an alternative wireless technology in which sensors are inductively powered by a wireless reader[25,31]. Because NFC-based sensors can be battery-free, low-cost, and secure against eavesdropping, this technology has emerged as a versatile platform for developing skin-mounted or implanted electronics capable of measuring heart rate, tissue oxygenation, sweat electrolyte composition, ultraviolet light exposure, and other physiological parameters[32–39]. A major limitation of NFC technology, however, is that the sensors can function only within the near-field of the reader, which is at most a few centimetres for a mobile reader such as a smartphone[25,31]. This constraint has so far limited the use of wireless, battery-free sensors for continuous monitoring during exercise and other forms of unconstrained physical activity. Although prior studies have demonstrated multiple battery-free sensors for temperature/pressure mapping[40] and neonatal vital signs monitoring[41], the systems relied on large NFC readers integrated into bedding and is incompatible with use during upright activity. Recent work also demonstrated vital sign and body motion monitoring with multiple stretchable passive tags, but required clothing to incorporate an individually-powered readout circuit above each sensor[42].

Here, we demonstrate continuous physiological monitoring with a battery-free body sensor network using near-field-enabled clothing to establish wireless power and data connectivity around the human body. Specifically, we use low-cost conductive threads and computer-controlled embroidery to integrate ordinary clothing with near-field-responsive inductor patterns that are capable of wirelessly connecting multiple skin-mounted sensors to a reader separated by up to a metre in distance through proximity to the functional patterns. In contrast with prior efforts to integrate NFC functionality into textiles[43], the near-field-enabled clothing are entirely fabric-based and robust to daily wear because they do not incorporate fragile silicon integrated circuits or require connectors to interact with nearby devices. We develop textile designs that are compatible with NFC-enabled smartphones and devices without any modification, and demonstrate their use in enabling spinal posture monitoring and continuous measurement of temperature and gait during exercise with multiple wireless, battery-free sensors.

## Results

**Near-field-enabled clothing**. Near-field-enabled clothing enable continuous physiological monitoring with battery-free sensors by wirelessly connecting NFC-based devices at arbitrary positions on the body to a wireless reader (Fig. 1a). The clothing incorporate connected planar inductor patterns that function as relays between physically separated locations, fabricated by embroidering conductive thread on conventional textiles (see Methods section). When a wireless reader is placed in close proximity to such an inductor pattern (hub), a time-varying magnetic field (13.56 MHz industrial, scientific, and medical band) induces current throughout the relay (Fig. 1b, c), generating magnetic "hotspots" simultaneously at the terminal ends of the relay that enable wireless powering and connectivity with sensors otherwise beyond the range of conventional NFC (Fig. 1d, e, Supplementary Fig. 1). Whereas standard NFC allows a sensor-reader separation of at most a few centimetres (typically <4 cm for mobile devices), near-field relays enable operation up to a metre apart provided that the sensor and reader are in close proximity to the patterned clothing.

Figure 1f illustrates the function of a 1 metre long near-field relay for an NFC system in which the reader and sensor are modelled by concentric 3.1 cm diameter coils. As a measure of the wireless interaction strength, we evaluate the wireless power transfer efficiency $\eta$ under ideal impedance matching conditions, which is a function of only the quality factor of the reader inductor $Q_1$, quality factor of the sensor inductor $Q_2$, and the coupling coefficient $k$ (see Methods section). Full-wave simulations show that $\eta$ falls from ~70% in the concentric configuration to 1% when the sensor is laterally displaced by 5 cm without the relay. Placing the reader 1 cm above the hub of the relay, however, creates a magnetic hotspot at the terminal end of the relay 1 m away in which the field amplitude is increased by about five orders of magnitude (Supplementary Fig. 2), enabling $\eta$ to reach a maximum of 30% in the region, where previously no connectivity can be established (Fig. 1f). Within 1.5 cm lateral distance from the centre of the hotspot, $\eta$ remains above 10%. For comparison, the minimum efficiency required for reliable energy and data transfer can be estimated to be about 2%, considering an output power of 200 mW from the reader and power consumption of 4 mW at the sensor node (see Methods section). Experimental measurements using a smartphone-sized NFC antenna and an NFC sensor validate the connectivity over the 1 m long relay, with measured efficiencies $\eta$ exceeding 6% within 1.5 cm proximity of the relay (Supplementary Fig. 3). As with conventional near-field technologies, the vertical operating distance from the inductive patterns is limited to a few centimetres distance (Supplementary Fig. 4), which can provide physical security against eavesdropping attacks in contrast to radiation-based wireless technologies such as Bluetooth.

**Design and characterisation**. We designed and fabricated near-field-enabled clothing capable of establishing near-field connectivity across wrists and torso with a long-sleeved shirt (Fig. 2a). The connectivity of this design was validated by placing a mobile, battery-powered NFC reader above the central hub over the chest and four sensor nodes configured with light-emitting diodes (LEDs) within 1 cm from the terminals. Operation of the reader at

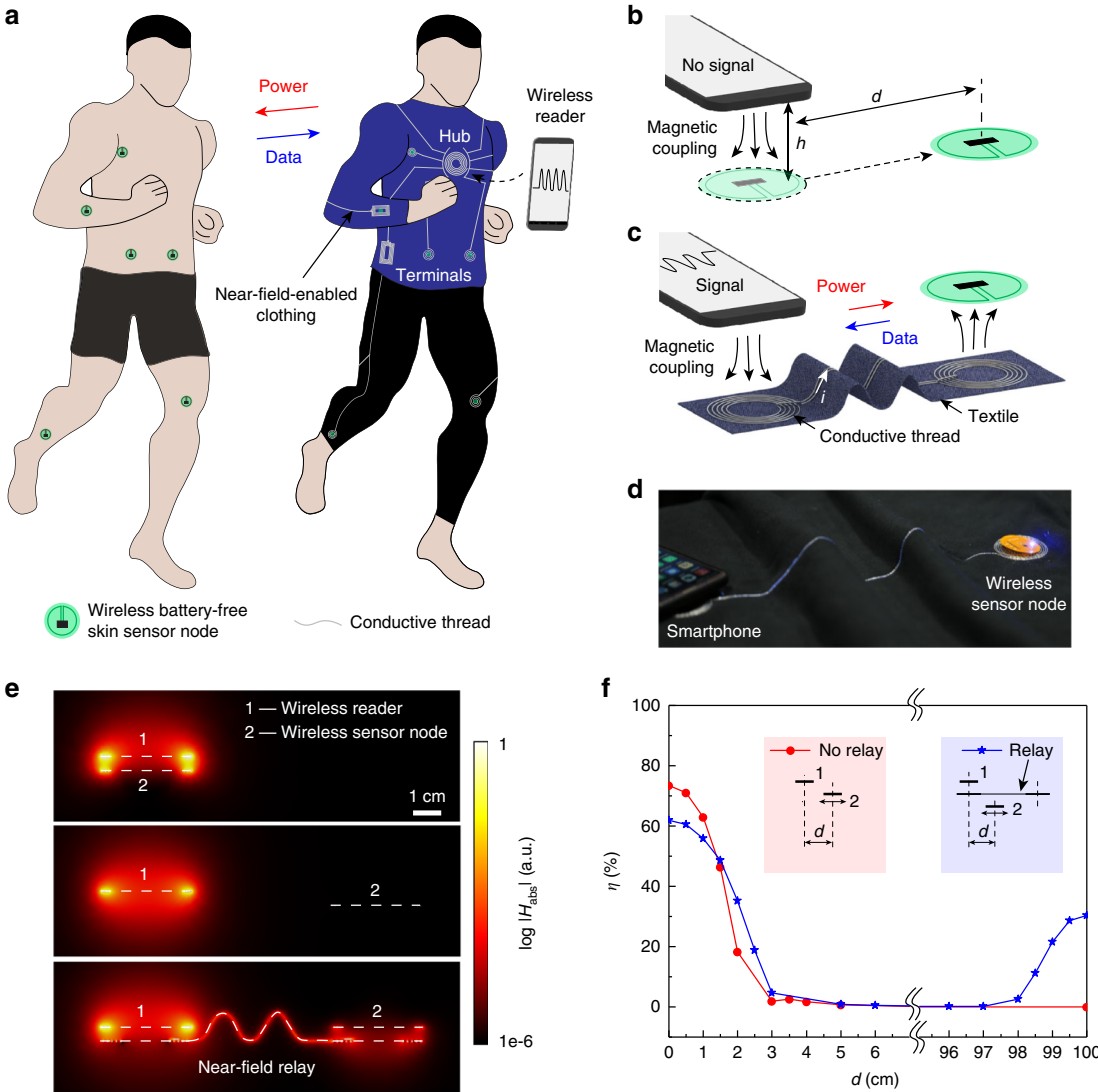

**Fig. 1 Battery-free sensor networks based on near-field-enabled clothing. a** Illustration of multiple battery-free sensor nodes mounted on the skin and interconnected to a wireless reader through the near-field-enabled clothing. **b, c** Conventional near-field communication (**b**) is limited to at most a few centimetres separation between the reader and sensor, while near-field relays (**c**) enable near-field connectivity upto metre scale in separation. $h$ is vertical distance, $d$ lateral distance, and $i$ current. **d** Photograph of a smartphone wirelessly powering a sensor node over a relay (40 cm length). **e** Normalised magnetic field $|H_{abs}|$ generated by a reader directly above a sensor, reader offset from sensor, and reader interconnected to sensor by a near-field relay. Reader and sensor diameters are 3.1 cm and distances $h = 0.5$ cm and $d = 8$ cm. **f** Power transfer efficiency $\eta$ as a function of sensor position $d$ relative to the reader without (red) and with (blue) a relay (1 m length). $\eta < 1\%$ in the concatenated region.

an output power of 200 mW results in robust activation of LEDs located near terminals at a distance of up to 1 m (from the central hub to wrist terminal, Supplementary Table 1), even during human motion (Supplementary Movie 1, Supplementary Fig. 5).

To design the patterns on the clothing, we developed a design process that combines circuit modelling and full-wave simulations to optimise $\eta$ as a function of the parameters shown in Fig. 2b (see Methods section). Owing to the long wavelength relative to the length scale of the relays, the configuration can be accurately modelled by the circuit in Fig. 2c even when the interconnects between the inductors are meandered or structure is placed on a nonplanar surface. The design procedure yields an inductor pattern optimised for interconnecting multiple 3.1 cm diameter reader/sensors with radius $r = 1.7$ cm, turn number $N = 10$, and wire gap $g = 1$ mm (Supplementary Fig. 6) at an operating frequency of 13.56 MHz. Using conductive thread (silver-plated polyamide yarn) with conductivity $\sigma = 7.8 \times 10^4$ S m$^{-1}$, the near-field relay enables the efficiency to be maintained at 30% between

length $l = 10$ cm to 1 m (Fig. 2d), decreasing to 6.3% when $l = 3$ m due to ohmic losses in the thread and transmission line effects (Supplementary Fig. 7).

Near-field-enabled clothing are fabricated using computer-controlled embroidery to integrate conductive thread on clothing with programmable patterns (Supplementary Fig. 8). The conductive thread, which is encapsulated in insulating transparent thermoplastic polyurethane, is highly suitable for daily wear since its electrical performance is robust (<5% variation) to repeated bending (minimum 1 mm radius) over 5000 cycles or immersion in 70 °C hot water for over 200 h (Supplementary Fig. 9). Although the thread is only slightly stretchable, patterns with serpentine design achieve stretchability of up to 100% (Supplementary Fig. 10) while maintaining the same transmission efficiency (Fig. 2e).

Near-field relays can be tailored to mitigate the sensitivity of NFC connectivity to the alignment of the sensor and reader with the clothing. By varying the inductor patterns, the magnetic

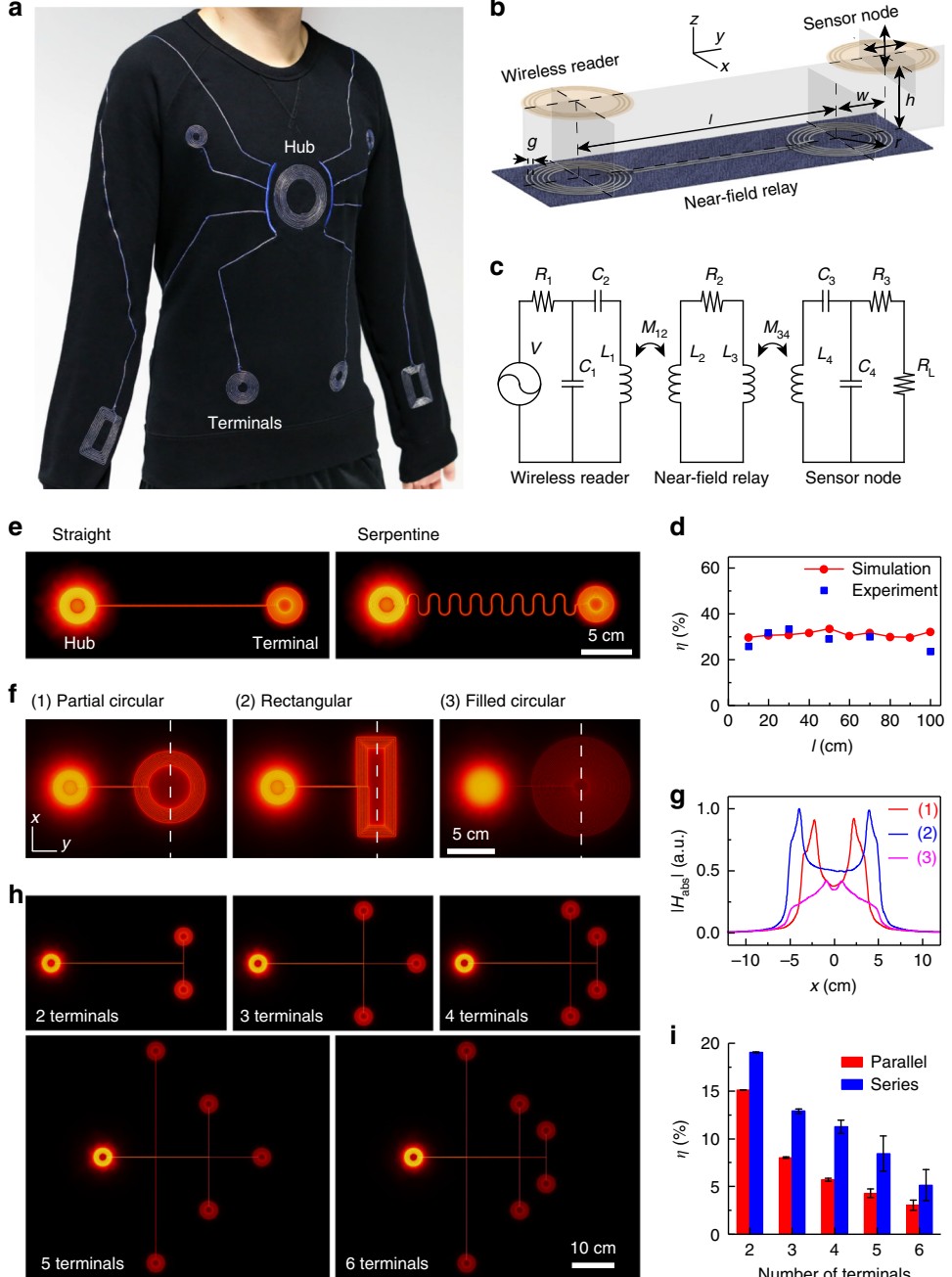

**Fig. 2 Near-field-enabled clothing design and characterisation. a** Photograph of near-field-enabled long-sleeved shirt comprised of a network with a single hub and eight terminals. **b** Configuration of the near-field relay when interconnecting a wireless reader with a sensor node. **c** Circuit model of the relay system in **b**. **d** Power transfer efficiency $\eta$ as a function of relay length $l$. The geometrical parameters are $r = 1.7$ cm, $N = 10$, $g = 0.1$ cm, $h = 1$ cm, and $w = 0$ cm. **e** Normalised magnetic field amplitude $|H_{abs}|$ above relays with straight and serpentine interconnects. **f** Magnetic field generated by designs (1) partial circular (7.2 cm diameter), (2) rectangular (10 cm length, 4 cm width) and (3) filled circular (10 cm diameter, 70% area filling ratio). Designs (1) and (2) have a area filling ratio of ~38%. **g** Magnetic field profiles along the dashed white lines in **f**. **h** Magnetic field above relays with multiple terminals connected in parallel. $l = 30$ cm between the hub and each terminal. **i** Power transfer efficiency $\eta$ as a function of the total number of terminals connected either in parallel or in series. Error bars show mean ± s.d. ($n$ = number of terminals).

field distribution can be controlled to achieve an operational area and/or working distance not achievable by the reader alone. Full-wave simulations show that increasing the radius $r$ while changing the number of turns $N$ and the wire gap $g$ allows sensors to operate over an increased 5 cm diameter region (Supplementary Fig. 11–13). Figure 2f, g show the magnetic field distribution of representative inductor designs optimised for vertical range $h$ (partial circular), directional displacement $w$

(rectangular), or operational area (filled circular) (Supplementary Fig. 14). As with conventional near-field systems, the performance is sensitive to the orientation of the devices. Measurement of the efficiency $\eta$ on the angular misalignment $\theta$ between the sensor and an inductive pattern through a 1 m long relay indicates that NFC connection can be maintained up to $\theta = 60°$ at which $\eta$ decreases by about threefold (Supplementary Fig. 15). This sensitivity is partially mitigated by the conformity

of the clothing to the body, which maintains angular alignment between the inductive patterns and skin-mounted sensors.

Simultaneous readout of multiple sensors can also be achieved using near-field relays with multiple inductor patterns. When interconnected either in series or parallel (Fig. 2h and Supplementary Fig. 16), excitation of a single terminal by a reader results in the creation of multiple magnetic hotspots that can be exploited for multinodal sensor operation. Although series interconnection results in constant current through all the inductors and higher efficiencies, parallel interconnection is more robust because it maintains function even if an inductor pattern is disconnected. For both the series and parallel networks, the transfer efficiency $\eta$ is inversely proportional to the number of terminals due to the distribution of energy over multiple terminal points (Fig. 2i). For a parallel relay network of 6 terminals, the transfer efficiency to each sensor node is measured to be about 3%, which is sufficient to establish reliable communication at an output power of 200 mW from the reader.

We investigated the scaling of system performance with number of terminals using a commercially available NFC reader that implements anti-collision protocols supported by current NFC standards (Supplementary Fig. 17, Methods). When maintaining a constant output power (200 mW antenna output), the distribution of energy to multiple terminals is manifested as a decrease in the maximum distance above the terminal at which each sensor can be read out, varying from 4.5 cm for a single terminal to 1.5 cm for six terminals (Supplementary Fig. 17b). The data rate also scales inversely proportional to the number of terminals: the rate at which each sensor is sampled decreases from 8 Hz for a single sensor to about 1.3 Hz for six sensors (Supplementary Fig. 17c). The total power consumption of the reader does not depend on the number of terminals or sensors as the output power from the antenna is constant (Supplementary Fig. 17d).

Near-field relays are also highly robust to deformation and wetting. When the relay is placed conformally to a curved surface, the efficiency varies by less than 2% with the interconnecting wires (or both inductor and sensor node) subjected to surface with 2 mm (5 mm) radius, while the efficiency is reduced by 20% with only the inductor pattern subjected to surface with 2 mm radius due to redistribution of the magnetic field (Supplementary Fig. 18a–c). Wetting of either the inductor patterns or the interconnecting wires results in less than 3% change in efficiency, indicating the robustness of the wireless connection against weather or sweat (Supplementary Fig. 18d). Two near-field relays can also be wirelessly interconnected by placing their inductor patterns in close proximity (Supplementary Fig. 19), which could be exploited to transmit energy and data between different articles of clothing.

**Multi-node spinal posture monitoring**. Real-time monitoring of spinal posture outside of clinical or laboratory settings has important applications in the diagnosis and treatment of musculoskeletal disorders such as neck and back pain, radiculopathy, and sensorimotor deficit[44–46]. We demonstrate continuous sensing of spinal posture by interconnecting multiple battery-free strain sensors distributed along the cervical, thoracic, and lumbar sections of the spine with a near-field-enabled clothing (Fig. 3a). Each skin-mountable sensor combines a commercial NFC chipset with a custom resistive elastomeric strain gauge, which exhibits up to 100% strain and a gauge factor of 3 (Fig. 3b, Supplementary Fig. 20a). Application of a stepwise strain profile over minute-length time scales relevant to posture sensing demonstrates accurate wireless strain measurement compared to direct wired readout of the strain sensor. The strain sensor exhibits a sufficiently rapid response (up to 2 Hz, 50% strain, Supplementary

Fig. 20b) to capture fast body motions, although additional circuitry and/or signal processing[47] may be required to resolve sensor hysteresis arising primarily from the viscoelastic effects of the elastomeric substrate. Owing to their battery-free operation, each skin-mountable sensor is lightweight (~0.3 g) and flexible, reducing the mechanical load on the user.

A multi-node near-field relay embroidered on a polyester-spandex shirt wirelessly interconnects the sensors to an NFC reader worn on the right arm above the relay hub (Supplementary Fig. 21, Supplementary Table 1). The near-field relay incorporates rectangular spiral inductor patterns for maintaining a robust power transfer efficiency from the reader to the sensor of 23% within a ≤3 cm displacement of the sensor from the centre-aligned position (Supplementary Fig. 14b). Wirelessly acquired sensor data did not depend on the terminal position (<3% variation) since the measurements are digitised with 14-bit resolution prior to transmission, and no detectable latency in data transmission was observed since the current flow through the relay is nearly instantaneous relative to the data rate (Supplementary Fig. 22). The near-field-enabled clothing also provides high-fidelity monitoring from skin-mountable sensors during moderate displacement between the skin/sensor and clothing layer.

Figure 3c shows continuous strain measurements acquired by the sensor network at 1 Hz sampling rate during physiological motions. Simultaneous data acquisition from the sensors, which are uniquely identified using standard NFC multiplexing protocols (Supplementary Fig. 21, Methods), enabled three distinct motions to be distinguished: neck motion results in a relative resistance change $\Delta R/R_0$ (maximum 150%) for S1 but not S2 or S3, lower back motion results in signals for S2 and S3 (maximum 60%) but not S1, and whole spine motion results in signals for all three sensors. Consistent recording across five repetitions of each motion (~10 s duration) demonstrates the reliability of the wireless connection (Supplementary Movie 2). We further validated the wireless strain measurements against a camera-based motion capture system with wearable reflective markers, which is considered as a gold standard in posture detection (see Methods section). The wirelessly acquired sensor resistance and the camera-based estimate of the cervical spine bending angle were in excellent agreement during three cycles of neck motion (Supplementary Fig. 23). In contrast with the motion capture system, however, the sensor network is not constrained by the field-of-view of the camera system, and can provide continuous monitoring of posture during daily activity for applications such as real-time corrective posture feedback.

**Continuous exercise monitoring**. Wireless battery-free sensors could provide important monitoring capabilities for athletic and health monitoring, but continuous operation of NFC-based devices during exercise has not previously demonstrated due to challenges in maintaining connectivity with the wireless reader. We overcome this challenge by using near-field-enabled clothing to interconnect NFC-based temperature and strain sensors to a smartphone reader during untethered running (Fig. 4a, b). The resulting monitoring system enables real-time measurement of axillary temperature, an important marker of health and performance[48], and running gait, an indicator for exhaustion and certain neurodegenerative diseases[49–51], using a temperature sensor node under the armpit and a strain sensor node on the knee, respectively (Supplementary Figs. 24, 25, Supplementary Movie 3). Although the sensor nodes are not directly accessible to the reader during physiological motion, continuous connectivity is maintained by near-field relays embroidered on the polyester-spandex shirt and pants (Fig. 4c, d, Supplementary Table 1).

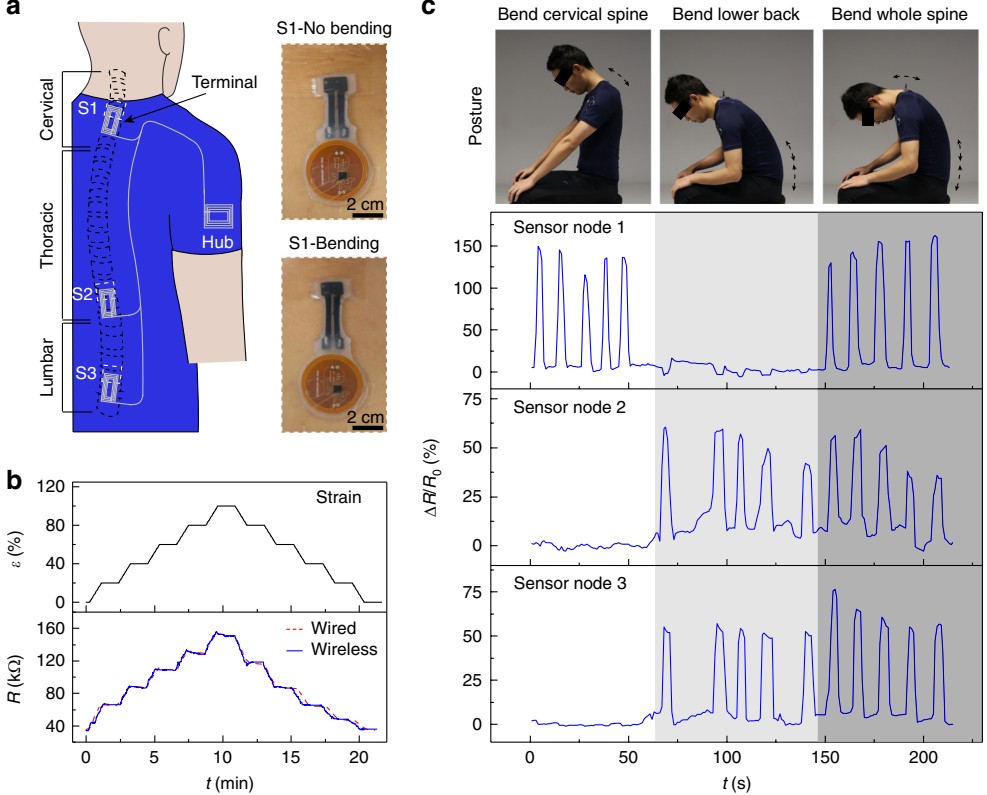

**Fig. 3 Multi-node spinal posture monitoring with battery-free sensors. a** Schematic of a wireless sensor system for real-time posture monitoring. Battery-free strain sensors mounted on cervical (S1), thoracic (S2), and lumbar (S3) sections are simultaneously interconnected to an NFC hub device via the near-field-enabled clothing. Photographs show example of S1 without/with bending cervical spine. **b** Profile of applied strain $\varepsilon$ and measured sensor resistance $R$ acquired through wired connection and wireless NFC. $\varepsilon$ is cycled from 0 to 100% with a step size of 20%. **c** Sensor data acquired during human subject motion. The subject sequentially bends the cervical spine, lower back, and whole spine, repeating each motion for five times.

We used a battery-free temperature sensor node to measure axillary temperature. Calibration measurements with a wired thermocouple show close agreement with the temperature sensor node (Fig. 4e). Figure 4f shows temperature measurements from a healthy subject during acclimatisation from indoor to outdoor tropical climate (28 °C, 70% relative humidity) in a thermal chamber, walking on a treadmill with stepwise increase in speed, and walking on a treadmill at constant speed (Fig. 4a, see Methods section). As expected, the sensor detected a rapid 1.5 °C increase in axillary temperature over ~6 min of the acclimatisation phase, followed by a slow 0.5 °C increase during walking, attributed to an increase in skin blood flow during physiological activity. Walking at a constant speed resulted in small (<0.5 °C) fluctuations in body temperature caused by perspiration and evaporative cooling. Data were acquired in real-time at a 4 Hz sampling rate without any loss of connectivity over the 40-min duration of the protocol, despite wetting of the clothing due to perspiration. This continuous mode of temperature monitoring could enable detection of exertional heat stress during athletic activity, military carriage, and other environments unsuited for conventional monitoring technologies (such as infrared imaging)[52].

Near-field relays embroidered on running pants enabled strain measurements to be simultaneously acquired from a battery-free sensor mounted on the knee. Figure 4g shows representative data during the walking phase compared to angular velocity measurements obtained from a battery-powered gyroscope attached to the ankle (Supplementary Fig. 26). The stride time $\Delta t$ calculated from both signals using a peak detection algorithm show close agreement (<1.5% difference) throughout the variable speed

phase (Fig. 4h), indicating that the strain sensor provides a reliable gait measurement. The data can be used to compute the stride time variability (STV), a clinically relevant marker for neurodegenerative disorders[49,53], for applications in performance feedback and intervention. When the treadmill speed $v$ is increased from 2.8 to 9 km h$^{-1}$, the corresponding decrease in STV (2.1–1.74%) is less than the decrease in $\Delta t$ (0.98–0.65 s) (Fig. 4i), which is an indicator of the fitness of the subject[49–51]. For young healthy adults, STV should be less than 3% over a long duration of exercise[49,53]. Measurements by both the battery-free strain sensor and the gyroscope show that STV remains below this threshold (2.58% and 1.86%, respectively) over the remaining duration of the exercise protocol (30 min) (Fig. 4j, k).

## Discussion

We have demonstrated continuous physiological monitoring with battery-free sensors during exercise using near-field-enabled clothing. These clothing integrate electromagnetically responsive patterns that extend the connectivity of near-field technologies from a range of few centimetres around a wireless reader to metre-scale networks of multiple battery-free sensors in proximity to these patterns. Simulations and circuit models illustrate principles and procedures for designing these patterns on clothing, and examples of posture and exercise monitoring at a trail system level demonstrate continuous wireless powering and sensing in unconstrained environments.

Current solutions for wireless sensor interconnection either require each sensor to be independently powered (such as with batteries)[11,17–23] or have a very limited range of operation (few centimetres)[32–39]. Networks of skin-mountable sensors

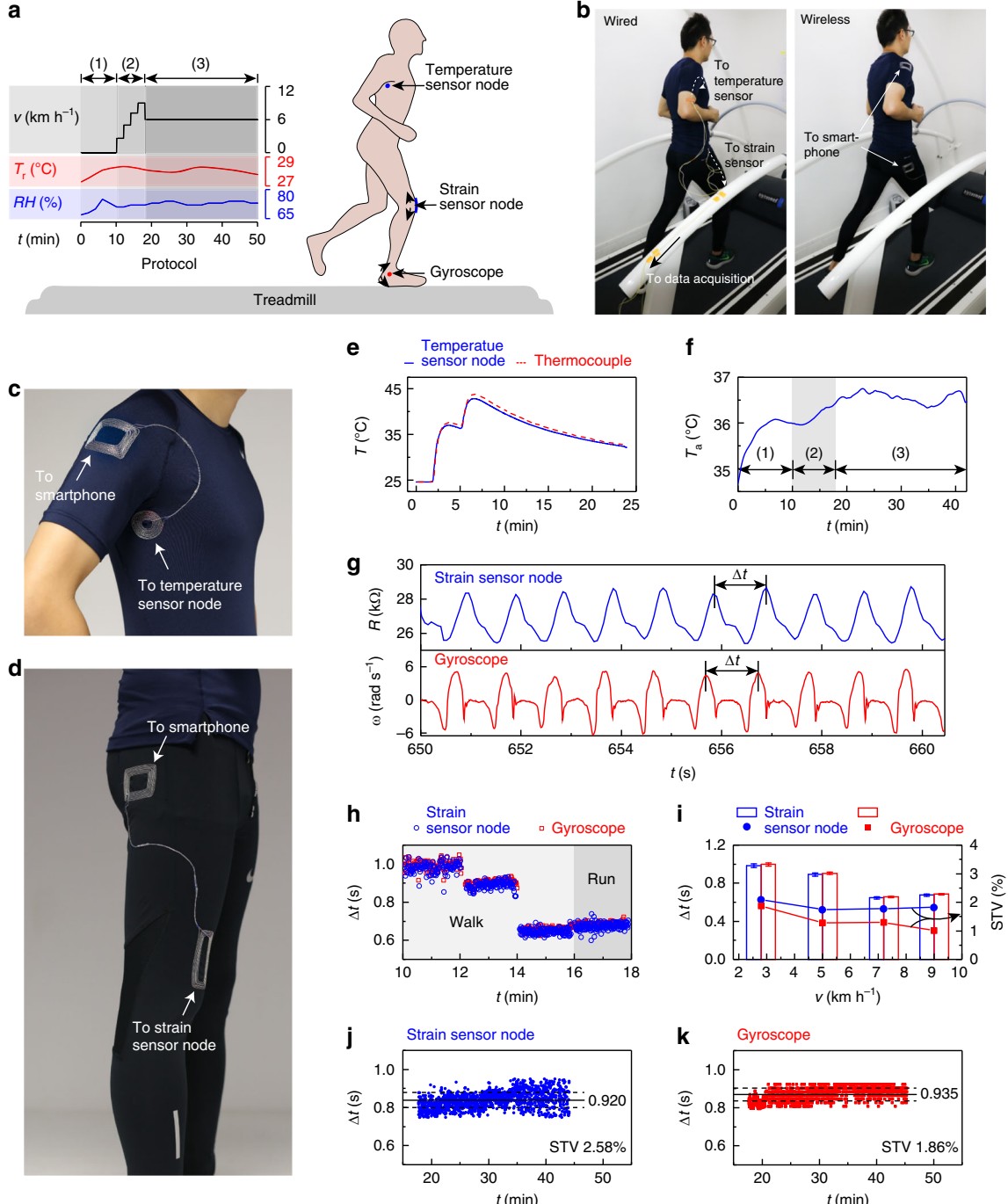

**Fig. 4 Continuous exercise monitoring with battery-free sensors. a** Experimental protocol and sensor placement. The subject performs (1) sitting, (2) walking/running at 2.8, 5, 7.2, and 9 km h$^{-1}$, and (3) walking at 6 km h$^{-1}$ in a thermal chamber (28 °C, 70% relative humidity). Battery-free sensors provide temperature and strain measurements when interconnected to a smartphone. A gyroscope attached at the ankle is used as reference for gait monitoring. **b** Comparison of a wired and wireless monitoring system. **c, d** Images of near-field-enabled shirt (**c**) and pants (**d**) for sensor interconnection. **e** Battery-free temperature sensor node calibration compared to wired thermocouple. **f** Axillary temperature $T_a$ recorded by the smartphone during the exercise protocol. **g** Representative strain sensor and gyroscope measurements during walking at 2.8 km h$^{-1}$. $\Delta t$, stride time given by the peak-to-peak interval. **h** $\Delta t$ measured by the strain sensor node and gyroscope during phase (2). **i** Comparison of $\Delta t$ and stride time variability (STV) acquired by the strain sensor node and gyroscope. Error bars show mean ± s.d. ($n = 120$ measurements). **j, k** $\Delta t$ and STV measured by the strain sensor node (**j**) and gyroscope (**k**) during phase (3).

interconnected by near-field-enabled clothing can therefore open engineering and clinical opportunities infeasible with present approaches. By eliminating rigid and bulky batteries, for example, wireless sensors can be lightweight and physically imperceptible on the skin, enabling improved signal quality and user comfort.

These sensors do not need to be removed or recharged and may be left on the body for extended periods of time, provided that a single power source at the hub (such as the smartphone) is available. Their low cost also allows single-use operation in which devices are sterilised, deployed on the body, and disposed after

use. The ability to perform real-time readout of such sensors by a single wearable device allows full-body monitoring for both clinical applications in rehabilitation (physical therapy), radiological imaging, assisted living, as well as athletic applications in performance assessment and fatigue detection.

Compared to conventional electronic textiles in which sensors and circuits are directly incorporated into clothing[42,54,55], near-field-enabled clothing derive their function from their passive electromagnetic response and are therefore entirely comprised of fabric. They are free of active electronic components that may be vulnerable to washing and daily wear, and interact wirelessly with nearby devices without the need for the user to plug or unplug connectors. Because the clothing directly manipulates the spatial distribution of the magnetic field around the body, the clothing is compatible with any NFC-enabled device, including smartphones, without modification. The safety of such devices is addressed by limitations in radio-frequency exposure specified by commercial standards, which remain valid when used with near-field-enabled clothing provided that the output power is unchanged.

Realising clinical systems based on such wireless sensor networks will require advances in sensor functionality and validation. In particular, the integration of additional sensing modalities, such as electrophysiology, pulse oximetry, respiration, and sweat analysis, or larger numbers of sensors may be required to reliably detect pathologies, which in turn pose increased technical requirements in the data rates and multiplexing capabilities of the wireless system. The reliability of each sensor node also needs to be validated over its expected lifetime, which may range from a few hours to several days for disposable devices. Sensors based on unconventional material sets and integration strategies can also yield more lightweight and conformable devices, although the conductivity of such materials will need to be improved to implement robust wireless operation. In addition, the user experience may be enhanced through aesthetic modifications of the near-field-enabled clothing, such as the use of coloured conductive threads for invisible integration or selective pattern display.

## Methods

**Numerical simulation**. Electromagnetic simulations were carried out with CST Microwave Studio (Dassault Systems). Field distributions excited by a 3.1 cm diameter coil at 13.56 MHz were calculated using the frequency domain solver. Simulation model of wireless reader (transmitter, power source) and sensor node (receiver) were used the coil in commercial NFC tag (TIDM-RF430-TEMPSENSE, Texas Instruments), which is made of copper and has 3.1 cm in outer diameter. The electrical conductivity of the conductive thread was $7.8 \times 10^4 \, \text{S m}^{-1}$ based on experimental measurement. The spacing between the wireless reader/sensor nodes and near-field relay was set at 1 cm unless otherwise stated.

**Fabrication of near-field-enabled clothing**. The near-field relays were designed using commercial software (PE-DESIGN 10, Brother) and uploaded to a sewing/embroidery machine (NV180, Brother). Conductive thread (235/36 dtex 2-ply HC + B TPU, Shieldex) was loaded into the machine and embroidered onto polyester-spandex or cotton-polyester shirts by computer-control. Connection of conductive threads was done using conductive epoxy (CW2460, Chemtronics) and sealed using hot-melt glue (DREMEL).

**Wireless measurement**. Scattering parameters between the wireless reader and sensor node with/without the near-field relay were measured using a vector network analyser (PicoVNA 106, Pico Technology) and coaxial cables (SMA–SMA, 50 Ω, Amphenol). The wireless reader and sensor node used the copper coil in commercial NFC tag (TIDM-RF430-TEMPSENSE, Texas Instrument). Under the condition of no relay, the vertical spacing between the reader and sensor node was fixed at 1 cm. Under the condition of using relay, the vertical spacing between the reader/sensor node and relay was fixed at 1 cm. For investigating misalignment effect, the sensor node connected to the coaxial cable was clamped on a translational stage and moved against the wireless reader/relay.

**Power transfer efficiency calculation**. The power transfer efficiency $\eta$ under ideal impedance matching conditions was calculated as[31,56]

$$\eta = \frac{k^2 Q_1 Q_2}{\left(1 + \sqrt{1 + k^2 Q_1 Q_2}\right)^2}, \tag{1}$$

where $k$ is the coupling coefficient between the reader and sensor inductors, and $Q_n$ the quality factor of the reader (labelled 1) and sensor (labelled 2) resonators. Measurement of $\eta$ was performed by obtaining the impedance matrix elements $Z_{nm}$ using a two-port vector network analyzer and calculating $k = \frac{\text{Im}(Z_{21})}{\sqrt{\text{Im}(Z_{11}) \, \text{Im}(Z_{22})}}$ and $Q_n = \frac{\text{Im}(Z_{nn})}{\text{Re}(Z_{nn})}$.

**Characterisation of conductive thread**. The bending test used a conductive thread that was clamped at both ends by a linear stage. The distance between the two ends was varied from 10 to 5 mm, which bends the thread to a minimum radius of ~1 mm. The thread was bent repeatedly for 5000 cycles and its electrical resistance was recorded by a sourcemeter (Keithley 2450, Tektronix).

The tensile test used conductive threads that were integrated on a polyester-spandex clothing with a straight/serpentine pattern. The conductive threads were clamped at both ends by a linear stage and stretched from its original length of 5 to 10 cm. Both the pulling force and electrical resistance were measured during tensile testing.

**Strain sensor node fabrication**. The strain sensor node was composed of a strain sensor and a commercial NFC tag. Strain sensors were fabricated starting from coating a 1:1 (base:agent) weight ratio mixture of Ecoflex$^{\text{TM}}$ 00-50 (Smooth-On) on 200 μm thick Polyethylene terephthalate (PET) films by a film applicator (TMAX-XT-200CA). The mixture was cured at room temperature overnight and became a thin film with ~200 μm thickness. Carbon black (BLACK PEARLS 2000, Cabot Corporation) was then manually rubbed on the Ecoflex$^{\text{TM}}$ 00-50 film covered with a PET mask, which was patterned by laser-cut (Universal Laser Systems, VLS 2.30). After removing excess carbon black and the mask, the strain sensor was connected to two leading wires at both ends using galinstan, and sealed by the other layer of Ecoflex$^{\text{TM}}$ 00-50. The strain sensor was connected to the commercial NFC tag (Supplementary Fig. 27a), termed as strain sensor node. The whole strain sensor node was finally encapsulated in Ecoflex$^{\text{TM}}$ 00-50.

There are two circuits for connecting the strain sensor with the NFC tag (Supplementary Fig. 27b, c). Circuit one was designed for high sampling rate (16 Hz) while circuit two was designed for high accuracy but low sampling rate (<10 Hz). In circuit one (voltage divider, Supplementary Fig. 27b), a constant input voltage $V_i = 2.2$ V was applied through the strain sensor $R_s$ and a reference resistor $R_r$, the output voltage $V_o$ was measured and used to calculate $R_s = \frac{(V_i - V_o)}{V_o} R_r$. In circuit two (Supplementary Fig. 27c), current $I \approx 2.4$ μA were flowed through two independent resistors $R_r$ and $R_s$, the corresponding output voltages $V_r$ and $V_s$ were measured and used to calculate $R_s = \frac{V_s}{V_r} R_r$. The frontend circuit for the strain sensor has a power consumption of 24.2 μW at the maximum resistance of 200 kΩ.

**Calibration of temperature/strain sensor node**. The temperature sensor node was a commercial NFC tag with an integrated NTC thermistor (ERTJ1VS104A, Panasonic Co. Ltd.). The frontend circuit for the temperature sensor has a power consumption 1.15 μW, while the total power consumption for the NFC chip is less than 4 mW in the active state as specified by the RF430FRL15xH family technical reference manual. The thermometer (HH506RA, Omega) connected with a thermocouple (K-type) was used as reference. During calibration, the closely-placed (<1 mm distance) thermistor and thermocouple were simultaneously heated up by a hotplate and cooled down by air.

The strain sensor node was clamped to a linear stage, and the NFC reader was held closely to the sensor node for powering and collecting data. The sensor node was cycled from strain between 0 and 100% with a step of 20%. The strain sensor was then disconnected from the NFC tag, directly wired to a digit multimeter (34461A, KEYSIGHT), and calibrated by the same procedure.

A camera-based motion capture system (VICON Motion Systems Ltd., UK) was used to provide gold-standard measurements for the cervical spine angle during bending motion. Three 1 cm-diameter reflective markers were attached to the head, cervical spine and thoracic spine as shown in Supplementary Fig. 23a. The strain sensor was attached adjacent to the reflective marker on the cervical spine and connected to the wireless NFC chip placed on the back region to prevent line-of-sight obstruction of the marker. Movie recording was performed using 6 infrared cameras, which enabled continuous recording of marker positions at a 200 Hz sampling rate. The cervical spine angle $\theta_s$ was computed using the three-dimensional coordinates of the markers, using the middle marker as the joint.

**Multi-node spine posture monitoring**. Spine posture monitoring was performed on a human subject wearing an athletic shirt integrated with a near-field relay. The relay was composed of one rectangular hub (6 cm × 4 cm dimension, 10 turns and 1.5 mm wire gap) at the sleeve and three rectangular terminals (10 cm × 4 cm

dimension, 8 turns and 1.5 mm wire gap) at the back. Three terminals were connected in parallel. Three strain sensor nodes were attached by transparent film dressings (Nexcare, 3M) at cervical, thoracic and lumbar spines, and right under each terminal. The commercial NFC reader (TRF7970A EVM, Texas Instruments) was placed in the pocket sewed on top of the hub and used to wirelessly power the three sensor nodes and collect data simultaneously.

The reader performed the multi-sensor readout using the anti-collision protocol specified by the ISO 15639 standard and implemented by a microcontroller (MSP430F2370). The anti-collision sequence is based on the slotted ALOHA protocol in which the reader broadcasts the number of slots and a mask value to each sensor. Each sensor responds at the time slot matching its masked unique identifier. If collision is detected, a new mask value is calculated, and the sequence is repeated until there are no collisions. The number of slots used in the implementation is 16.

**Continuous exercise monitoring**. Continuous exercise monitoring included axillary temperature sensing and gait monitoring. The experiment was carried out on a human subject wearing an athletic shirt and pants, both integrated with a near-field relay. The relay for temperature sensing was composed of a rectangular hub (6.8 cm × 5.4 cm dimension, 10 turns and 1.5 mm wire gap) at the sleeve and a circular terminal (2.1 cm outer radius, 10 turns and 1.5 mm wire gap) at the armpit. The relay for gait monitoring was composed of a rectangular hub (6.8 cm × 5.4 cm dimension, 10 turns and 1.5 mm wire gap) at the back pocket and a rectangular terminal (10 cm × 4 cm dimension, 8 turns and 1.5 mm wire gap) at the knee. An NFC-enabled smartphone was placed in the pocket right on the hub and the wireless battery-free sensor node was attached to the armpit/knee by the transparent film dressing. A 9-axis motion sensor (MOVESENSE) including acceleration sensor, gyroscope, magnetometer, and Bluetooth was attached to the ankle. The angular velocity measured from the gyroscope (104 Hz sampling rate) was used as reference for gait monitoring.

The experiment was carried out in a thermal chamber with 28 °C ambient temperature and 70% relative humidity to mimic warm and humid conditions. The protocol was composed of three stages. At stage (1), the human subject seated on a chair for 10 min to get accustomed to the environment and to derive a baseline skin temperature. At stage (2), the human subject walked/ran on a treadmill (h/p/comos) with gradually increasing speeds from 2.8, 5, 7.2 to 9 km h$^{-1}$, and maintained each speed for 2 min. At stage (3), the human subject walked on the treadmill at a constant speed of 6 km h$^{-1}$ for 30 min.

All experiments complied with guidelines by the National University of Singapore. All subjects were volunteers and provided informed consent.

## Data availability

All data supporting this study and its findings within the article and its Supplementary Information are available from the corresponding authors upon reasonable request.

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

## Acknowledgements

The authors thank Z. Goh for assistance with the art in Figs. 1–2. J.S.H. acknowledges support from the National Research Foundation Singapore (NRFF2017-07 and AISG-GC-2019-002), Ministry of Education Singapore (MOE2016-T3-1-004), and Institute for Health Innovation and Technology grants.

## Author contributions

R.L. and J.S.H. conceived and planned the research. R.L., H.J.K., S.A., S.A.K., S.C.C.T., H.Y., B.C.K.T. and J.S.H. designed the near-field-enabled clothing and wireless battery-free sensor nodes, and performed the experiments. S.C.C.T. and J.K.W.L. designed the human lab trials. R.L. and J.S.H. wrote the paper with input from all the authors.

## Competing interests

The authors declare no competing interests.
