## [Peer Review File · Nature Communications]

Reviewers' comments:

Reviewer #1 (Remarks to the Author):

General Comments

This article developed wireless battery-free body sensor networks using near-field-enabled clothing, achieving conformal, power delivery, and long-range measurement at the system level. The wireless system and near-field-enabled clothing characteristics were systematically studied, including near-field simulation, power efficiency in each sensor node, and clinical tests. In summary, with a few major adjustments in claims, the paper will make an impressive contribution to the field of wireless electronics.

Major Comments

1. Currently, clothes combined with RFID and NFC technology have been commercialized and the results of a study on fabric electronics (DOI: 10.1002/adma.201900564) are being reported. Therefore, it is necessary to discuss which developed system have novelty compared to them. Also, the word 'meter-scale' causes confusion. It feels like it has solved a short distance of communication, which is a disadvantage of NFC technology. Therefore, further explanation of length extension in the same plane should be added.
2. In the FIG. 3. b, in strain sensor, the recovery resistance is different according to each strain and appear to take a lot of time (~5 mins). Human motion is relatively fast, and it is necessary to discuss whether the developed strain sensor is suitable for measuring the body's strain and hysteresis problem of the carbon black based composite.
3. What is the operating power of the temperature and strain sensor? The power of a smartphone is very small and should be discussed whether the distributed power of each node is sufficient to operate each sensor.
4. Depending on human movement, the angle between the antenna attached to the clothing and the sensor node changes. As a result, the power that is transferred will be different and resistance of sensors will also be different. We need systematic study data for this.
5. Individuals' performance of the commercial temperature and developed strain sensors are different. Therefore, calibration is required for use in the same platform and performance comparison with other commercial references is required (strain sensor).
6. The authors should better highlight that they currently have achieved a trial system level with human's exercise, rather than full clinical test studies. For a clinical test study, much more data are required, discussing reliability and lifetime issues of the sensors.
7. Lastly, it is necessary to show several sensors-operated videos at the hub position (FIG. 1. a) of the clothing at the same time.

Reviewer #3 (Remarks to the Author):

In this paper, the authors present a near-field-enabled clothing system to interconnect multi-node sensors distributed at distances of up to a meter around the body to a wireless readout device (smartphone) through near-field relays (conductive threads on clothing). Each terminal reader antenna wirelessly acquires data from each nearby skin-mounted sensor. A central hub antenna, wired to each terminal reader antenna, wirelessly provides measurements to a smartphone using

NFC multiplexing protocols. The work extends the lateral distance between each battery-free sensor and a smartphone, but still requires close proximity (short vertical distance) between the tag and the reader antennas (i.e. between sensors and terminal antennas, and between a central hub and a smartphone). The concepts are interesting, and they represent useful extensions of recently reported schemes for addressing multiple body-mounted sensors using long-range NFC antennas. Nevertheless, the authors must describe in quantitative detail whether the near-field relays for sensing multiple distant nodes limit the maximum vertical distance between the tag and the reader antennas, and the maximum data transfer rate of NFC -- power consumption is another, critically important issue in the practical utility of this scheme, as the battery for power supply must also be carried on the body. These issues, as well as those listed below, must be discussed before the paper can be accepted.

1. Standard NFC allows tag-to-reader communication typically within 4 cm for mobile devices. The authors should include quantitative information on wireless power transfer efficiency (η) as a function of separation distance between the smartphone to the sensor (as shown in Fig. 1e) when placing the smartphone 1-4 cm above the hub of the relay.
2. On page 5, it is mentioned that an η of 10% is sufficient to perform reliable energy and data transfer via NFC. The authors should explain how η of 10% is defined to be sufficient. Fig. 2i shows that η falls below 10% for a network with a single hub and three terminals in parallel. To achieve reliable 1-m (or 30-cm) near-field communication, how many terminals can interconnect to a single hub antenna?
3. Fig. 3 : Authors provide sensor data acquired from a network when placing a hub closer to S1 than S2 and S3. In Fig. 3c, S1 data is about twice as large as S2 and S3 data. Checking the latency or attenuation through the near-field relay is recommended. Authors could place two sets of sensors (three on left back, other three on right back) to be symmetrical about the spine, and one hub of right-back sensors (R1-3) is placed close to R1 (right top on back as same as in Fig. 3a) and one hub of left-back sensors (L1-3) close to L3 (left bottom on back).
4. Fig. 3 and 4: The authors should clarify the distance between a hub and each sensor. It will be important to demonstrate continuous multi-node monitoring up to meter scale as stated in this work.
5. Authors used a 1- or 4-Hz sampling rate during testing. What is the maximum data transfer rate of the system with a single hub and N (i.e. N=3 or 8) terminals for a 1-m (or 30-cm) near-field communication? The low sampling rates demonstrated by the authors greatly restrict the range of possible applications.

November 4, 2019

Re: Nature Communications manuscript NCOMMS-19-28170, Lin, et al “Wireless battery-free body sensor networks using near-field-enabled clothing”

To the reviewers,

We thank all the reviewers for their insightful and constructive comments on our manuscript. Please find below a point-by-point response to each reviewer. The specific changes made to the manuscript to address each point are highlighted in blue.

Reviewer #1

This article developed wireless battery-free body sensor networks using near-field-enabled clothing, achieving conformal, power delivery, and long-range measurement at the system level. The wireless system and near-field-enabled clothing characteristics were systematically studied, including near-field simulation, power efficiency in each sensor node, and clinical tests. In summary, with a few major adjustments in claims, the paper will make an impressive contribution to the field of wireless electronics.

We thank the reviewer for affirming the potential interest of this manuscript to the field of wireless electronics. As detailed below, we have carefully adjusted the claims in the manuscript to address the specific points raised.

1. Currently, clothes combined with RFID and NFC technology have been commercialized and the results of a study on fabric electronics (DOI: 10.1002/adma.201900564) are being reported. Therefore, it is necessary to discuss which developed system have novelty compared to them. Also, the word 'meter-scale' causes confusion. It feels like it has solved a short distance of communication, which is a disadvantage of NFC technology. Therefore, further explanation of length extension in the same plane should be added.

We thank the reviewer for pointing out the need to compare our approach with prior work integrating NFC/RFIC technology with textiles. We would like to emphasize that the focus of our work is distinct from the conventional “electronic textiles” approach, which seeks to integrate sophisticated devices such as organic transistors into fabrics, in that we aim to wirelessly interconnect NFC devices using only conductive threads. This stands in contrast with prior work^[R1] where the RFID/NFC chip and other electronic components needs to be “wired” into clothing. Our clothing can therefore be robust to daily wear because they consist of entirely fabric and do not need to be physically connected to fragile electronic components. In addition, our clothing can power and communicate with skin electronics without physical connection, ensuring their robust and intimate monitoring of human physiological signal. We have added the suggested reference and revised both the Introduction (Page 4) and Discussion (Page 11-12) sections to clarify the above points:

“In contrast with prior efforts to integrate NFC functionality into textiles, the near-field-enabled clothing are entirely fabric-based and robust to daily wear because they do not incorporate fragile silicon integrated circuits or require connectors to interact with nearby devices.” (Page 4)

Compared to conventional electronic textiles in which sensors and circuits are directly incorporated into clothing, near-field-enabled clothing derive their function from their passive electromagnetic response and are therefore entirely comprised of fabric. They are free of active electronic components that may be vulnerable to washing and daily wear, and interact wirelessly with nearby devices without the need for the user to plug or unplug connectors. Because the clothing directly manipulates the spatial distribution of the magnetic field around the body, the clothing is compatible with any NFC-enabled device, including smartphones, without any modification. The safety of such devices is addressed by limitations in radio-frequency exposure specified by commercial standards, which remain valid when used with near-field-enabled clothing provided that the output power is unchanged.” (Page 11-12)

We also thank the reviewer for pointing out the potential confusion regarding the term “meter-scale”. We have removed the term from the Abstract and Introduction. We use the term once in the Discussion section, but only in the context of a “meter-scale network” where there should be no confusion with the distance of free-space wireless communication. As requested, we have further revised the Abstract (Page 2) and Results section (Page 5) to clarify the need to be in proximity to the clothing:

“Here, we report near-field-enabled clothing capable of establishing wireless power and data connectivity between multiple distant points around the body to create a network of battery-free sensors interconnected by proximity to functional textile patterns.” (Page 2).

“Whereas standard NFC allows a sensor-reader separation of at most a few centimetres (typically <4 cm for mobile devices), near-field relays enable operation up to a meter apart provided that the sensor and reader are in close proximity to the patterned clothing.” (Page 5)

Also, to support that the relays can be “meter-scale” in length, we have added new supplementary table (Supplementary Table 1) providing the distances between the hubs and terminals for each relay network. The relay network shown in Fig. 2a is “meter-scale” as each of the sleeves are 1 m from the central hub; the functionality of this network is demonstrated in a supplementary figure and video (Supplementary Fig. 5a,b, Supplementary Video 1).

[R1] Cui, L., Zhang, Z., Gao, N., Meng, Z., & Li, Z. (2019). Radio Frequency Identification and Sensing Techniques and Their Applications—A Review of the State-of-the-Art. *Sensors*, 19(18), 4012.

2. In the FIG. 3. b, in strain sensor, the recovery resistance is different according to each strain and appear to take a lot of time (~5 mins). Human motion is relatively fast, and it is necessary to discuss whether the developed strain sensor is suitable for measuring the body's strain and hysteresis problem of the carbon black based composite.

We thank the reviewer for raising these important questions regarding the performance of strain sensor. We note that the applied strain in Fig. 3b is intended to cover the time scales typically encountered in spinal posture sensing, and was therefore programmed to change relatively slowly (over minutes). To clarify this point, we have revised Fig. 3b to include the applied strain profile. We have also conducted additional experiments to test the response of the sensor at smaller time scales. No significant distortion in the sensor response was found for cyclical application of 50% strain at 2 Hz, which should be sufficient for most human motions (Supplementary Fig. 20b). For spinal motion, the strain encountered is not expected to exceed 50% (Ref. R2 and Fig. 3c).

We agree with the reviewer that our strain sensor exhibits hysteresis, which is experimentally characterized in Supplementary Fig. 20a. Although the hysteresis is significant, we note that it is comparable to state-of-

the-art strain sensors with similar stretchability (about 20% in Ref. R3-R4) because it originates from the viscoelastic nature of polymer substrate and its interface with the conductive material^[R3,4]. This hysteresis does not prevent the sensor from being used in clinical sensing applications because it can be compensated by using additional circuits and software when higher precision becomes necessary^[R5].

We have included additional discussion of the above points in the Results section (Page 8):

“Application of a stepwise strain profile over minute-length time scales relevant to posture sensing demonstrate accurate wireless strain measurement compared to direct wired readout of the strain sensor. The strain sensor exhibits a sufficiently rapid response (up to 2 Hz, 50% strain, Supplementary Fig. 20b) to capture fast body motions, although additional circuitry and/or signal processing may be required to resolve sensor hysteresis arising primarily from the viscoelastic effects from the elastomeric substrate (Supplementary Fig. 20a).”

Fig.3 Multi-node spinal posture monitoring with battery-free sensors. **a**, Schematic of a wireless sensor system for real-time posture monitoring. Battery-free strain sensors mounted on cervical (S1), thoracic (S2), and lumbar (S3) sections are simultaneously interconnected to an NFC hub device via the near-field-enabled clothing. **b**, Profile of applied strain ϵ (top) and measured sensor resistance R (bottom) acquired through wired connection and wireless NFC. ϵ is cycled from 0 to 100% with a step size of 20%. **c**, Sensor data acquired during human subject motion. The subject sequentially bends the cervical spine, lower back and whole spine, repeating each motion five times.

Supplementary Figure 20. Calibration of strain sensors. a, Dependence of relative resistance change $\Delta R/R_0$ on strain ε cycled from 0 to 100% with a step of 20%. The same strain sensor is connected in a wired approach (to a digit multimeter) or wireless approach (to a NFC tag). **b**, Response of a strain sensor to $\varepsilon=50\%$ at applied frequency of 0.5Hz, 1Hz and 2Hz.

- [R2] Ianuzzi, A., & Khalsa, P. S. (2005). Comparison of human lumbar facet joint capsule strains during simulated high-velocity, low-amplitude spinal manipulation versus physiological motions. *The Spine Journal*, 5(3), 277-290.
- [R3] Amjadi, M., Kyung, K. U., Park, I., & Sitti, M. (2016). Stretchable, skin-mountable, and wearable strain sensors and their potential applications: a review. *Advanced Functional Materials*, 26(11), 1678-1698.
- [R4] Qiu, A., Li, P., Yang, Z., Yao, Y., Lee, I., & Ma, J. (2019). A Path Beyond Metal and Silicon: Polymer/Nanomaterial Composites for Stretchable Strain Sensors. *Advanced Functional Materials*, 29(17), 1806306.
- [R5] Wang, X., & Ye, M. (2008). Hysteresis and nonlinearity compensation of relative humidity sensor using support vector machines. *Sensors and Actuators B: chemical*, 129(1), 274-284.

3. What is the operating power of the temperature and strain sensor? The power of a smartphone is very small and should be discussed whether the distributed power of each node is sufficient to operate each sensor.

We thank the reviewer for raising important points regarding the power consumption of the sensor and readout device. We note that the both the strain and temperature sensors are passive resistive sensors and therefore do not directly consume any power. The sensor readout is performed using a voltage divider circuit (Supplementary Figure 27) that interfaces with the ADC of a commercial NFC chip (TI, RF430FRL152h). We have calculated the power consumption of the sensor frontend to be about 24.2 μW and 1.15 μW for the strain and temperature sensors respectively and added this data to Methods section. This requirement is very small compared to that of the NFC chip, which in the active mode is consumes < 4 mW, mostly by the integrated microcontroller^[R6,R7].

As noted by the reviewer, the power consumption of the wireless system is dominated by the readout device. The output power of a NFC-enabled smartphone is difficult to exactly determine because it is generally proprietary and varies by manufacturer, but is expected to be in the range of 100 to 500 mW^[R6]. This power requirement is comparable to that of cellular communication and is within the capability of smartphones (indeed most advanced smartphones are already NFC-enabled). For comparison, the custom NFC reader shown in Supplementary Fig. 17 has an output power of 200 mW and in our experiments achieves a slightly larger range than the smartphone used in Fig. 4. We emphasize that our sensor networks use the same output power as in standard NFC – the enhanced connectivity is a result of the electromagnetic properties of the

clothing and not increased power consumption. We note that near-field-enabled clothing can provide ~3% power transfer efficiency to 6 sensor nodes in parallel connection (Fig. 2i), which is more than sufficient to operate all the sensors based on these conservative estimates.

To clarify these points, we have revised the Results section (Page 5 and 7).

“For comparison, the minimum efficiency required for reliable energy and data transfer can be estimated to be about 2%, considering an output power of 200 mW from the reader and sensor node power consumption of 4 mW (Methods).” (Page 5)

“For a parallel relay network of 6 terminals, the transfer efficiency to each sensor node is measured to be about 3 %, which is sufficient to establish reliable communication at an output power of 200mW from the reader.” (Page 7)

[R6] “RF430FRL15xH Family Technical Reference Manual,” 2014.

[R7] Zhao, Y., Smith, J. R., & Sample, A. (2015, April). NFC-WISP: A sensing and computationally enhanced near-field RFID platform. *In 2015 IEEE International Conference on RFID (RFID)* (pp. 174-181). IEEE.

4. Depending on human movement, the angle between the antenna attached to the clothing and the sensor node changes. As a result, the power that is transferred will be different and resistance of sensors will also be different. We need systematic study data for this.

We thank the reviewer for pointing out the critical issue of sensor-reader alignment. *As requested by the reviewer, we carried out systematic study on the dependence of the power transfer efficiency η on the angular misalignment θ between the sensor node and the terminal of relay (Supplementary Fig. 15).*

The angular misalignment affects η by changing magnetic coupling. The slight enhancement of η by increasing θ to 10° arises because the sensor inductor is brought closer to the relay during rotation, and then decays beyond 20° due to reduction in the normal component of magnetic flux through the inductor. The NFC reader becomes unable to connect with the sensor node at $\theta > 60^\circ$. We note that such extreme angular misalignment is unlikely in the case of skin-mounted sensors because the clothing naturally conforms with the body. The angular misalignment also has a minimal effect on the measured sensor resistance as the value is digitized prior to wireless transmission through the NFC protocol. *We have added a discussion on the angular misalignment effect in the Results section (Page 7).*

“As with conventional near-field systems, the performance is sensitive to the orientation of the devices. Measurement of the efficiency η on the angular misalignment θ between the sensor and an inductive pattern through a 1-m long relay indicate that NFC connection can be maintained up to $\theta = 60^\circ$ at which η decreases by about threefold (Supplementary Fig. 15). This sensitivity is partially mitigated by the conformity of the clothing to the body, which maintains angular alignment between the inductive patterns and skin-mounted sensors.”

Supplementary Figure 15. Effect of angular misalignment on near field communication. **a**, Illustration of power transferring from a reader tag to a sensor node via a 1-m long relay. The reader and the sensor node are placed 1cm and 2cm above the relay, respectively. **b**, Dependence of power transfer efficiency η on the angular misalignment θ between the sensor node and the relay. There is no connection between the NFC reader and the sensor node when $\theta > 60^\circ$.

5. Individuals' performance of the commercial temperature and developed strain sensors are different. Therefore, calibration is required for use in the same platform and performance comparison with other commercial references is required (strain sensor).

We thank the reviewer for the raising the need for sensor calibration to commercial standards. As noted by the reviewer, the temperature sensor is commercial – we therefore do not calibrate this sensor. We would like to point out that our custom strain sensor is calibrated using a commercial mechanical testing instrument (NLE series linear stage, Newmark) in which a precisely known strain is applied and the corresponding resistance measured (Fig. 3b and Supplementary Fig. 20). The use of such a calibration instrument should obviate the need to compare with commercial strain sensors, of which there are currently none available in the required 50% strain range.

We agree with reviewer that comparing the performance of the complete sensor platform with a commercial standard would be insightful. To address this request, we simultaneously acquired posture data using both the platform described in Fig. 3 and a camera-based motion capture system (VICON Motion Systems), which is considered a gold standard for posture detection. The results show excellent agreement between the measured cervical spine angle θ_s (based on displacement of three reflective markers) and the wireless sensor measurement ΔR (Supplementary Fig. 23), which points to the potential of our platform to enable continuous posture monitoring in environments where camera tracking is impractical.

In addition to the above supplementary figures, we have also revised the Results section (Page 9) to discuss the calibration results.

“We further validated the wireless strain measurements against a camera-based motion capture system with wearable reflective markers, which is considered a gold standard in posture detection (see Methods). The wirelessly acquired sensor resistance and the camera-based estimate of the cervical spine bending angle were in excellent agreement during three cycles of neck motion (Supplementary Fig. 23). In contrast with the motion capture system, however, the sensor network is unconstrained by line-of-sight to a camera, and can provide continuous monitoring of posture during daily activity for applications such as real-time corrective feedback of harmful posture.”

The details of motion capture experiment are described in the revised Methods section (Page 15):

“Comparison with motion capture system. A camera-based motion capture system (VICON Motion Systems Ltd., UK) was used to provide gold-standard measurements for the cervical spine angle during bending motion. Three 1 cm-diameter reflective markers were attached to the head, cervical spine and thoracic spine as shown in Supplementary Fig. 23a. The strain sensor was attached adjacent to the reflective marker on the cervical spine connected to the wireless NFC chip placed on the back region to prevent line-of-sight obstruction of the marker. Video recording was performed using 6 infrared cameras, which enabled continuous recording of marker positions at a 200 Hz sampling rate. The cervical spine angle θ_s was computed using the three-dimensional coordinates of the markers, using the middle marker as the joint.”

Supplementary Figure 23. Posture monitoring with battery-free sensor and camera-based motion capture. **a**, Photograph of the experimental setup. Three reflective markers along the cervical region of the spine are placed for the camera-based motion capture system (VICON Motion Systems). The strain sensor is attached to the skin next to the center marker, and the NFC circuit placed below the markers to avoid occlusion. **b**, Camera-based measurement of the cervical spine angle θ_s and battery-free sensor measurement ΔR during three continuous bend-and-release cycles. **c**, ΔR as a function of θ_s during bending and releasing of the cervical spine. Error bars show mean \pm s.d. ($n=3$ cycles).

6. *The authors should better highlight that they currently have achieved a trial system level with human's exercise, rather than full clinical test studies. For a clinical test study, much more data are required, discussing reliability and lifetime issues of the sensors.*

We fully agree with the reviewer that the current work demonstrates the concept of near-field-enabled clothing only at the level of unconstrained human exercise – the clinical validation of a system based on this concept remains an important direction for future work. We have carefully gone over the manuscript to ensure that there are no claims of “clinical validation” and have included additional discussion in the main text (Page 12) on the reliability and lifetime issues that will need to be addressed before such systems can be clinically.

“Realising clinical systems based on such wireless sensor networks will require advances in sensor functionality and validation. In particular, the integration of additional sensing modalities, such as electrophysiology, pulse oximetry, respiration, and sweat analysis, or larger numbers of sensors may be required to reliably detect pathologies, which in turn pose increased technical requirements in the data rates and multiplexing capabilities of the wireless system. The reliability of each sensor node also needs to be validated over its expected lifetime, which may range from a few hours to several days for disposable devices. Sensors based on unconventional material sets and integration strategies can also yield more lightweight and conformable devices, although the conductivity of such materials will need to be improved to implement robust wireless operation. In addition, the user experience may be enhanced through aesthetic modifications of the near-field-enabled clothing, such as the use of coloured conductive threads for invisible integration or selective pattern display.”

7. *Lastly, it is necessary to show several sensors-operated videos at the hub position (FIG. 1. a) of the clothing at the same time.*

We thank the reviewer for pointing out the need to demonstrate the specific network illustrated in Fig. 1a in which the hub is positioned over the chest. As requested, we have added a new video (Supplementary Video 1) demonstrating the use of the relay network shown in Fig. 1a consisting of a circular (10 cm diameter) hub at chest and eight terminals distributed around the body. The video shows that a single wearable NFC reader (powered by a small battery) placed above the chest can wirelessly power four sensor nodes at the same time, as indicated by activation of LEDs on each sensor, that is robust to human motion. The details of the network and the results are also shown in a new supplementary figure (Supplementary Fig. 5). These results are also discussed in the main text in the revised Results section (Page 5-6).

“The connectivity of this design was validated by placing a mobile, battery-powered NFC reader above the central hub over the chest and four sensor nodes configured with light-emitting diodes (LEDs) within 1 cm from the terminals. Operation of the reader at an output power of 200 mW results in robust activation of LEDs located near terminals at a distance of up to 1 m (from the central hub to wrist terminal, Supplementary Table 1), even during human motion (Supplementary Video 1, Supplementary Fig. 5).”

Supplementary Figure 5. Wireless powering of four LED nodes. **a**, Photograph of a near-field-enabled clothing. The clothing is composed of a hub located at chest and eight terminals distributed around body. Four wireless LED nodes are placed proximity to terminals. **b**, Photograph of wireless powering through the clothing. A battery-powered NFC reader (**c**), which is placed on the central hub, wirelessly powers four LED nodes (**d**) through the near-field-enabled clothing. The LED node is composed of a blue LED connected to a loop antenna with 3.1-cm diameter.

Reviewer #2

This work introduces a new interconnection of wearable sensors using the clothing and present its application for real-time, multi-node measurement of spinal posture and sensing of temperature and gait.

The work uses conductive threads creating near-field from inductor patterns that are capable of connecting multiple skin-connected sensors to a reader. I am working in the area of wearable sensors, and liked this interesting idea.

Novelty of the work is that through the conductive thread, multiple sensor points are connected through a hub location perhaps with a bigger inductive pattern with improved NFC distance and capability. Such a networked connection may present some interesting applications around the human body. Authors have already applied two scenarios where body movements/posture detection and exercise are detected.

We thank the reviewer for noting the interest of this work to the field of wearable sensors.

1. Simultaneous readout of multiple sensors is quite interesting. Authors please provide more information about how reliable is to read from multiple sensors. And what is the limit for how many sensors can be connected at once?

We thank the reviewer for raising these important questions. We would like to point out that we perform multi-sensor readout using a wearable, commercially available NFC reader (TRF7970A EVM, Texas Instruments, shown in Supplementary Fig. 5), which implements the anti-collision sequence specified by the ISO 15639 standard in order to acquire data from multiple sensors. The anti-collision sequence is based on the slotted ALOHA protocol in which collisions are handled by having each sensor retransmit in another generated time-slot (see revised Methods section). This protocol is also supported on the sensor end by the commercially available NFC chip (RF430FRL152h, Texas Instruments). The sensor readout therefore meets industry standards for reliability. The use of the near-field-enabled clothing does not affect this communication protocol, although it can enhance the ability of a single reader to electromagnetically couple to multiple distant sensors.

The maximum number of sensors allowed depends on the implementation of the NFC protocol. For the particular case of the TRF7970A EVM transponder, the maximum number of sensors is approximately limited by the number of slots (16 in this implementation) due to increasing collision rates^[R6]. The implementation may be modified to support a larger number of sensors.

We have added a new Methods section “Multi-sensor readout” to discuss these points (Page 15-16).

“Multi-sensor readout was performed using a wearable commercial NFC reader (TRF7970A EVM, Texas Instruments) powered by a small battery. The reader uses the anti-collision protocol specified by the ISO 15639 standard and implemented by a microcontroller (MSP430F2370). The anti-collision sequence is based on the slotted ALOHA protocol in which the reader broadcasts the number of slots and a mask value to each sensor. Each sensor responds at the time slot matching its masked unique identifier. If collision is detected, a new mask value is calculated, and the sequence is repeated until there are no collisions. The number of slots used in the implementation is 16.”

[R6] “Implementation of the ISO15693 Protocol in the TI TRF796x,” SLOA138, Texas Instruments, 2009.

2. Looking at Fig. 3a&c how do you know the data is collected is from S1 or S2 or S3? Please provide mechanism how you differentiate signals from multiple sensors.

As noted in the above response, the readout protocol for multiple sensors is specified by the ISO 15693 standard which uses the slotted ALOHA method for anti-collision during readout. This is implemented by a commercially available NFC reader and is used in this work without any modification.

We have revised the main text referencing Fig. 3c to direct readers to the relevant details (Page 9).

“Fig. 3c shows continuous strain measurements acquired by the sensor network at 1 Hz sampling rate during physiological motions. Simultaneous data acquisition from the sensors, which are uniquely identified using standard NFC multiplexing protocols (Supplementary Fig. 21, Methods)”

3. In conventional NFC, the reader should be in close proximity to a sensor to read data from sensor. Here the reader should also be placed near to the hub to read the data from sensors. Although the idea is interesting, please provide power/energy required for single and multiple sensor connections. Please provide some comparison.

We agree with the reviewer that the networks here require proximity between the reader and sensors with the near-field-enabled clothing. This stands in contrast with conventional NFC which requires direct physical proximity, and can therefore enable connectivity between distributed battery-free sensors that would otherwise be inaccessible. We emphasize that this enhanced connectivity is achieved using the same level of output power as in conventional NFC. **To study the power requirements, we performed an additional study directly monitored the total power consumption of the NFC reader using a USB power monitor, as shown in Supplementary Fig. 17.** The total power consumption P ranges from 0.555~0.556 W regardless of whether the near-field relays are used or the number of sensor nodes, which is consistent with the constant antenna output power of 200 mW. As also detailed in the response to Reviewer #3,

We have revised the Results section (Page 7) to discuss these important points.

“We investigated the scaling of system performance with number of terminals using a commercially available NFC reader that implements anti-collision protocols supported by current NFC standards (Supplementary Fig. 17, Methods). When maintaining a constant output power (200 mW antenna output), the distribution of energy to multiple terminals is manifested as a decrease in the maximum distance above the terminal at which each sensor can be read out, varying from 4.5 cm for a single terminal to 1.5 cm for 6 terminals (Supplementary Fig. 17b). The data rate also scales inversely proportional to the number of terminals: the rate at which each sensor is sampled decreases from 8 Hz for a single sensor to about 1.3 Hz for 6 sensors (Supplementary Fig. 17c). The total power consumption of the reader does not depend on the number of terminals or sensors as the output power from the antenna is constant (Supplementary Fig. 17d).

Supplementary Figure 17. Scaling of communication performance with number of terminals. **a**, Photograph of the experimental setup with a 1-terminal relay consisting of a NFC reader placed 1 cm above the hub and a temperature sensor placed distance h above the terminal. The relay is 30-cm long and the number of terminals in the parallel configuration increased as shown in Fig. 2h. **b**, Maximum vertical distance h_{\max} at which NFC connection can be established as a function of number of terminals. **c**, Sampling frequency f of each sensor by the reader as a function of number of terminals. Error bars show mean \pm s.d. ($n=100$ samples). **d**, Total power consumption P of the reader a function of number of terminals. The output power of the antenna is set to 200 mW.

4. *Quality of Fig. 1-e should be improved.*

We thank the reviewer for this suggestion and have carefully revised Fig. 1e (revised Fig. 1f) to improve its clarity. The goal of the Fig. 1f is to quantify the power transfer efficiency η achieved with and without the relay at the distance of 1 m. As such, we concatenate the efficiency plot to emphasize the performance near hub ($d=0$ to 7 cm) and terminal ($d=95$ to 100 cm) positions. We have also revised the caption to provide more detailed context to plot.

FIG. 1. Battery-free sensor networks based on near-field-enabled clothing. **a**, Illustration of multiple battery-free sensors mounted on the skin (left) and interconnected to a wireless device through the near-field-enabled clothing (right). **b-c**, Conventional near-field communication (**b**) is limited to at most a few centimetres separation between the reader and sensor, while near-field relays (**c**) enable near-field connectivity up meter scale in separation. h is vertical distance, d lateral distance, and i current. **d**, Photograph of a smartphone wirelessly powering a sensor node over a relay (40 cm length). **e**, Normalised magnetic field $|H_{\text{abs}}|$ generated by a reader directly above a sensor (top), reader offset from sensor (centre), and reader interconnected to sensor by a near-field relay (bottom). Reader and sensor diameters are 31mm and displacements $h = 5\text{mm}$ and $d = 80\text{mm}$. **f**, Power transfer efficiency η as a function of sensor position d relative to the reader without (red) and with (blue) a relay (1 m length). $\eta < 1\%$ in the concatenated region.

Reviewer #3

In this paper, the authors present a near-field-enabled clothing system to interconnect multi-node sensors distributed at distances of up to a meter around the body to a wireless readout device (smartphone) through near-field relays (conductive threads on clothing). Each terminal reader antenna wirelessly acquires data from each nearby skin-mounted sensor. A central hub antenna, wired to each terminal reader antenna, wirelessly provides measurements to a smartphone using NFC multiplexing protocols. The work extends the lateral distance between each battery-free sensor and a smartphone, but still requires close proximity (short vertical distance) between the tag and the reader antennas (i.e. between sensors and terminal antennas, and between a central hub and a smartphone). The concepts are interesting, and they represent useful extensions of recently reported schemes for addressing multiple body-mounted sensors using long-range NFC antennas.

We thank the reviewer for noting the interest and potential usefulness of this work.

Nevertheless, the authors must describe in quantitative detail whether the near-field relays for sensing multiple distant nodes limit the maximum vertical distance between the tag and the reader antennas, and the maximum data transfer rate of NFC -- power consumption is another, critically important issue in the practical utility of this scheme, as the battery for power supply must also be carried on the body. These issues, as well as those listed below, must be discussed before the paper can be accepted.

We would like to briefly highlight the additional studies performed to address to the reviewer's concerns regarding the scaling of readout distance with number of sensors, the maximum data transfer rate, and the power consumption. More detailed discussions can be found in response to the specific questions below.

- *Scaling of readout range with number of nodes.* We have added a new supplementary subfigure (Supplementary Fig.17b) to quantify the change in the vertical readout distance as the number of terminals on the near-field relay increases. The readout distance is quantified as h_{max} , which is the maximum sensor distance above the relay where NFC connection can be established with a reader placed at directly above a hub 30 cm away. The results show that h_{max} decreases from 4.5 cm to about 1.5 cm as the number of sensor terminals increases from 1 to 6, primarily due to the reduction in magnetic field flux as the current is distributed over more terminals (Fig.2h). We note that, in absence of the relay, the maximum readout distance between the sensor and reader is only about 7 cm at the same level of output power – the relay can significantly extend the range of NFC communication (up to 1 m) with minimal effect on the wireless readout distance.
- *Maximum data transfer rate.* We have added a new supplementary subfigure (Supplementary Fig. 17c) to quantify the change in data rate as the number of terminals increases. Specifically, we use a commercial NFC reader to communicate with $N=1$ to 6 temperature sensor nodes described in Supplementary Fig. 27, and measure the rate at which each sensor is sampled. The sampling rate is 8 Hz for a single sensor (this can be doubled to 16 Hz using an alternative, less sensitive frontend circuit), and decreases with the number of sensors as $1/N$ (also see response to Reviewer #2, Point 1).
- *Power consumption.* We have added a new supplementary subfigure (Supplementary Fig. 17d) to quantify the effect of terminal number on total power consumption P of the NFC reader. Using the same experimental setup as described above, we used a USB power meter to measured P , which is maintained at a constant 0.555~0.556 W regardless of the number of sensors or whether the relay is

used. This is consistent with the commercial NFC reader's specifications, which specifies a constant output power of 200 mW independent of the number of sensors detected or the strength of the connection.

1. Standard NFC allows tag-to-reader communication typically within 4 cm for mobile devices. The authors should include quantitative information on wireless power transfer efficiency (η) as a function of separation distance between the smartphone to the sensor (as shown in Fig. 1e) when placing the smartphone 1-4 cm above the hub of the relay.

We thank the reviewer for raising this point for clarification. As requested, we have added a new supplementary figure (Supplementary Fig. 3) that experimentally validates the dependence of η on the separation distance as shown in Fig. 1e (revised Fig.1f). The experimental setup consists of a custom smartphone NFC antenna (η cannot be directly measured using a smartphone), a NFC sensor node held 1 cm above the relay, and a 1-m long near-field relay. In absence of the relay, lateral displacement of the sensor by 5 cm results in a decrease in η from ~80% to ~4%, which is expected given that the range of conventional NFC is about 4 cm. With the relay, a magnetic hotspot is created at the terminal positioned 1 m from the reader, enabling NFC connectivity with $\eta \sim 20\%$ in the concentric position. Like conventional NFC, displacement of the sensor from the maximum position will also cause η to decrease: we measure that NFC connectivity can be within a 1.5-cm lateral distance where η exceeds 6% (considering an output power of 200 mW).

As suggested by the reviewer, we also repeat the measurement at the concentric position as the vertical distance between the NFC device and the relay varies from $h=0$ to 5 cm (Supplementary Fig. 3). η exceeds 4% within a 2.5-cm vertical distance. We also discuss these experimental results in the revised Results section (Page 5).

“Experimental measurements using a smartphone-sized NFC antenna and NFC sensor validate the connectivity over the 1-m long relay, with measured efficiencies η exceeding 6% at up to 1.5 cm proximity to the relay (Supplementary Fig. 3).”

Supplementary Figure 3. Power transfer efficiency from smartphone antenna to sensor node. a, Photograph of the experimental setup for measuring the power transfer efficiency η between a smartphone NFC antenna to a sensor node. The near-field relay is 1 m long. d is the lateral displacement of the sensor and h the vertical distance between the sensor and the relay. **b,** Photograph of the smartphone NFC antenna. The antenna consists of copper on a polyimide substrate and has dimensions $3.7 \text{ cm} \times 5.2 \text{ cm}$, wire width of 1mm, and wire gap of 0.3mm. **c,** η as a function of d without (in red color) and with (in blue color) the near-field relay for $h=1 \text{ cm}$. The plot is concatenated in the region $d=7$ to 95 cm where $\eta < 1\%$. **d,** η as a function of vertical distance h when the antenna and sensor are concentric to the hub and terminal of the relay.

2. On page 5, it is mentioned that an η of 10% is sufficient to perform reliable energy and data transfer via NFC. The authors should explain how η of 10% is defined to be sufficient. Fig. 2i shows that η falls below 10% for a network with a single hub and three terminals in parallel. To achieve reliable 1-m (or 30-cm) near-field communication, how many terminals can interconnect to a single hub antenna?

We thank the reviewer for raising this point for clarification. We would like to emphasize that the NFC readers used in this manuscript are commercially available, implement the standard NFC protocol (ISO 15693), and have the same output power with and without the relays. The efficiency η required to establish connectivity are therefore the same as in conventional NFC.

The efficiency requirements depend on specifics of the system, but can be estimated based on the output power of the NFC reader and the power consumption of the NFC tag. For the demonstrated system, the output power of the NFC reader is about 200 mW (typically varying from 100 to 500 mW for a mobile device), and power consumption of NFC sensor (including both the temperature and strain sensors) is less than 4 mW [R6,R7]. The efficiency of power transfer should therefore be larger than 2% in order to establish NFC connectivity, with additional margin to account for non-ideal impedance matching. Based on these estimates, the NFC reader should be able to establish reliable 30-cm communication with up to 6 sensor

nodes simultaneously (Fig. 2i). This is experimentally demonstrated and characterized in a new supplementary figure (Supplementary Fig. 17).

We have also revised the Results section (Page 5 and Page 7) to discuss these points:

“For comparison, the minimum efficiency required for reliable energy and data transfer can be estimated to be about 2%, considering an output power of 200 mW from the reader and sensor node power consumption of 4 mW (Methods).” (Page 5)

“For a parallel relay network of 6 terminals, the transfer efficiency to each sensor node is measured to be about 3 %, which is sufficient to establish reliable communication at an output power of 200mW from the reader.” (Page 7)

[R6] “RF430FRL15xH Family Technical Reference Manual,” 2014.

[R7] Zhao, Y., Smith, J. R., & Sample, A. (2015, April). NFC-WISP: A sensing and computationally enhanced near-field RFID platform. *In 2015 IEEE International Conference on RFID (RFID)* (pp. 174-181). IEEE.

3. Fig. 3: Authors provide sensor data acquired from a network when placing a hub closer to S1 than S2 and S3. In Fig. 3c, S1 data is about twice as large as S2 and S3 data. Checking the latency or attenuation through the near-field relay is recommended. Authors could place two sets of sensors (three on left back, other three on right back) to be symmetrical about the spine, and one hub of right-back sensors (R1-3) is placed close to R1 (right top on back as same as in Fig. 3a) and one hub of left-back sensors (L1-3) close to L3 (left bottom on back).

We thank the reviewer for raising the important question of latency and attenuation through the near-field relay. We would like to first clarify the reviewer’s observation that the measurement from S1 is about twice of S2 and S3 (Fig. 3c). We note that the measurements represent mechanical strain, and larger values of R indicates a larger strain, which is expected because the neck bends considerably more than the spine. Because these measurements are acquired using the ADC on the NFC chip and digitally transmitted over the NFC protocol, they are not directly related to the latency or attenuation of the NFC connection. We verify this behavior in a new supplementary figure (Supplementary Fig. 22b), which shows that the wireless strain sensor exhibits <3% variation with sensor position.

As suggested by the reviewer, we also measure the data transmission latency at each terminal position, defined as the time interval between sending the readout command and receiving the reply. The latency is ~44.5 ms regardless of which terminal the sensor is positioned (closer or farther to the reader) or even whether the relay is used (Supplementary Fig. 22c). The reason is because the magnetic field at the terminals is generated almost instantaneously (at the speed of current flow, which is near the speed of light) upon coupling of the reader to the hub. This time lag is negligible is compared to the clock rate of NFC chip, and as such the relay does not significantly impact the latency of the connection.

To make clarification, we add the following sentence to Results section (Page 8-9).

“Wirelessly acquired sensor data did not depend on the terminal position (<3% variation) since the measurements are digitized with 14-bit resolution prior to transmission, and no detectable latency in data transmission was observed as the current flow through the relay is nearly instantaneous relative to the data rate (Supplementary Fig. 22).”

Supplementary Figure 22. Variation in sensor readout from different terminals. **a**, Photograph of the experimental setup consisting of a NFC reader, a strain sensor node, and the relay network shown in Fig. 3 (hub and three terminals along the spinal column). **b-c**, Strain sensor resistance R (**b**) and latency of the wireless measurement (**c**) wirelessly measured at each terminal position. Terminal 0 represents close proximity wireless connection to the hub without the relay. Error bars represent mean \pm s.d. ($n=100$ measurements).

4. Fig. 3 and 4: The authors should clarify the distance between a hub and each sensor. It will be important to demonstrate continuous multi-node monitoring up to meter scale as stated in this work.

We thank the reviewer for raising this point for clarification. As requested, we have added new supplementary table (Supplementary Table 1) that provides the distances between the hubs and terminals for each relay network. The relay network shown in Fig. 2a is “meter-scale” as each of the sleeves are 1 m from the central hub. We have added a new supplementary figure and video to demonstrate the functionality of this network in wirelessly distributing power from a battery-powered NFC reader to four LED nodes (Supplementary Fig. 5a,b, Supplementary Video 1). The sensor networks used in Fig. 3-4 have a maximum sensor/hub distance of about 0.6 m; longer distances are easily feasible but were unnecessary in their particular context. We have also revised the Results section (Page 5-6) to discuss this point.

“The connectivity of this design was validated by placing a mobile, battery-powered NFC reader above the central hub over the chest and four sensor nodes configured with light-emitting diodes (LEDs) within 1 cm from the terminals. Operation of the reader at an output power of 200 mW results in robust activation of LEDs located near terminals at a distance of up to 1 m (from the central hub to wrist terminal, Supplementary Table 1), even during human motion (Supplementary Video 1, Supplementary Fig. 5).”

Supplementary Table 1. Dimensions of the sensor networks interconnected by various near-field-enabled clothing.

Network	Terminal position	Distance-to-hub* (cm)
Powering network (Fig. 2a)	Left sleeve	100
	Right sleeve	100
	Left abdominal	31.5
	Right abdominal	31.5
Spinal posture monitoring network (Fig. 3a)	Top	23.1
	Middle	54.3
	Bottom	57.2
Exercise monitoring network (Fig. 4c,d)	Temperature	30.1
	Gait	38.4

*Distance-to-hub is measured as the interconnect wire length between the inductive pattern of the hub and terminals.

5. Authors used a 1- or 4-Hz sampling rate during testing. What is the maximum data transfer rate of the system with a single hub and N (i.e. $N=3$ or 8) terminals for a 1-m (or 30-cm) near-field communication? The low sampling rates demonstrated by the authors greatly restrict the range of possible applications.

We thank the reviewer for raising this important question. The maximum sampling rate achieved in our system is 16 Hz (224 bits/s), which is limited by the communication protocol as implemented by the smartphone/commercial NFC reader and not the near-field relays. This sampling rate is sufficient for the temperature and gait/posture sensing demonstrations, although we agree that many other important applications, such as electrophysiology (ECG/EMG) or rapid motion detection, will require further improvement. The sampling rate scales as $1/N$ where N is the number of sensors, decreasing for example to 2.7 Hz for 3 sensors (Supplementary Fig.17c).

Prior work by the Rogers group^[R8] has demonstrated modified NFC systems (following standard ISO/IEC 15693) with sampling rates of up to 200 Hz and data transfer rates of up to 800 bytes/s for application on patients immobilized near a large readout coil. Such adaptations of the reader software and circuits should be applicable to networks based on near-field-enabled clothing as well. We have revised the Discussion section (Page 12) to highlight this important direction for future work.

“Realising clinical systems based on such wireless sensor networks will require advances in sensor functionality and validation. In particular, the integration of additional sensing modalities, such as electrophysiology, pulse oximetry, respiration, and sweat analysis, or larger numbers of sensors may be required to reliably detect pathologies, which in turn pose increased technical requirements in the data rates and multiplexing capabilities of the wireless system.”

[R8] Chung, H.U., Kim, B.H., Lee, J.Y., Lee, J., Xie, Z., Ibler, E.M., Lee, K., Banks, A., Jeong, J.Y., Kim, J. and Ogle, C., 2019. Binodal, wireless epidermal electronic systems with in-sensor analytics for neonatal intensive care. *Science*, 363(6430), p.eau0780.

REVIEWERS' COMMENTS:

Reviewer #1 (Remarks to the Author):

My overall opinion of this revision is as follows: The authors made sufficient revisions and the paper is suggested for publication.

1. The manuscript is significantly improved. It can be accepted without change.
2. My concerns have been addressed and the manuscript has been improved.
3. My concerns have been addressed and the manuscript has been improved.
4. My concerns have been addressed through additional experiment results.
5. My concerns have been addressed through additional experiment results.
6. The manuscript is significantly improved.
7. My concerns have been addressed through additional experiment results.

Reviewer #2 (Remarks to the Author):

I am happy with the details of responses authors provided to comments of reviewers. The manuscript looks a lot stronger now.
I congratulate authors for this nice work.

Reviewer #3 (Remarks to the Author):

The authors have carefully addressed all inputs from the referees. I feel that the revised versions is now suitable for publication.